# Research Progress and Improvement Ideas of Anti-Epidemic Resilience in China’s Urban Communities

**DOI:** 10.3390/ijerph192215293

**Published:** 2022-11-19

**Authors:** Peng Cui, Ping Zou, Xuan Ju, Yi Liu, Yalu Su

**Affiliations:** Department of Engineering Management, School of Civil Engineering, Nanjing Forestry University, Nanjing 210037, China

**Keywords:** urban communities, COVID-19, anti-epidemic resilience, dynamic network, computational experiments

## Abstract

In the post-epidemic era, China’s urban communities are at the forefront of implementing the whole chain of accurate epidemic prevention and control. However, the uncertainty of COVID-19, the loopholes in community management and people’s overly optimistic judgment of the epidemic have led to the frequent rebound of the epidemic and serious consequences. Existing studies have not yet formed a panoramic framework of community anti-epidemic work under the concept of resilience. Therefore, this article first summarizes the current research progress of resilient communities from three perspectives, including ideas and perspectives, theories and frameworks and methods and means, and summarizes the gap of the current research. Then, an innovative idea on the epidemic resilience of urban communities in China is put forward: (1) the evolution mechanism of community anti-epidemic resilience is described through the change law of dynamic networks; (2) the anti-epidemic resilience of urban communities is evaluated or predicted through the measurement criteria; (3) a simulation platform based on Multi-Agent and dynamic Bayesian networks simulates the interactive relationship between “epidemic disturbance–cost constraint-–epidemic resilience”; (4) the anti-epidemic strategies are output intelligently to provide community managers with decision-making opinions on community epidemic prevention and control.

## 1. Introduction

COVID-19 has been raging globally for more than two years since the beginning of December 2019. As of November 12, 2022, there were 638 million cumulative diagnoses and 6.62 million deaths worldwide (Data source: CDC; WHO; ECDC; Wikipedia; New York Times; China Health Commission). In order to effectively control the spread of COVID-19, countries around the world have taken effective measures. Spain declared a state of emergency and adopted a mandatory confinement policy [1]; Iran used continuous surveillance information to seek emergency relief from WHO to avoid further transmission [2]; Canada closed its borders to prevent offshore imports and South Africa implemented sanitary control measures to mitigate co-infection of COVID-19 and other infectious diseases [3], among others. At the same time, China has taken a series of measures: closing the exit routes from the city in areas where the outbreak is serious, urgently stopping large mass events, temporarily closing scenic venues, strictly enforcing health quarantine, etc. With the implementation of these measures, COVID-19 in China has been effectively controlled and stabilized, but as the virus continues to mutate, its lethality and infectious characteristics are erratic and unpredictable. With the end of the Spring Festival, the conclusion of the Winter Olympics and the arrival of all kinds of holidays, China’s epidemic prevention and control work is still facing greater pressure of “external prevention of importation and internal prevention of rebound”. In some areas, there have been reports of local outbreaks, and the risk of epidemic spread and spillover still exists.

As the end of the national governance system, the communities in China are not only the front line in the fight against the epidemic but also the coordinators between the government and the residents [4]. In February 2020, Xi Jinping emphasized that “China should sink the prevention and control forces to communities and strengthen the implementation of prevention and control measures, so that all communities become strong bastions to the epidemic”. Since then, China has issued a series of policies and documents related to community epidemic prevention and control (CEPC), such as the Notice on Strengthening CEPC of COVID-19 Infection, the Urgent notice on urban and rural communities to organize prevention and control of COVID-19 infection, the Notice on Further Improving the Precision and Refinement of CEPC, the Notice on the CEPC in urban and rural during the spring festival and so on. On 22 January 2022, at the press conference of the joint prevention and control mechanism of the State Council, community-related terms such as “community transmission” and “community prevention and control” were mentioned more than 50 times; for example: “Community residents are the key to standardizing CEPC”, “strengthen the construction of community public health committees”, “further organize and expand the residents to weave a community prevention and control network that is horizontal to the edge and vertical to the bottom”, “We issued the CEPC Manual to provide detailed operational guidance” and so on.

However, the epidemic outbreaks in communities are highly uncertain, which poses a dilemma for communities and governments in managing the epidemic in either the normal or emergency situations. On the one hand, comprehensive prevention under the normal state, with measures such as quarantine setting up, nucleic acid testing and vaccination, can reduce the risk of infection, but with high management and resource costs. In addition, it brings inconvenience to residents and is extremely detrimental to economic and social development. On the other hand, the disposal in the emergency state can be improvised, but it relies heavily on the daily accumulation of human and material resources. Otherwise, it is very likely to cause the epidemic and disposal costs to become out of control and thus bring new chain risks [5]. Numerous cases of community outbreaks demonstrate that many management deficiencies of CEPC exist in the normal and emergency work in China. For example: (1) redundant work, repeated display of health QR codes, temperature measurement, multiple registrations and reports; (2) the plight of vulnerable groups; over-reliance on information technology hinders the travel of the elderly; acute and critically ill patients have a difficult time seeking medical treatment during the epidemic; (3) formalism; irresponsible body temperature measurement; the statistical table prepared temporarily for the inspection of the superior; (4) insufficient protective measures, cross-infection caused by the irregular setting of testing points or chaotic on-site organization in the process of COVID-19 testing in the community; (5) information transmission blocked; community residents receive lagging or fake epidemic-related information; the community is suddenly closed, resulting in residents not making advance preparations; (6) one-size-fits-all control, using the epidemic as an excuse to shirk responsibilities, increasing the pressure upon subordinates and so on [6,7,8].

Therefore, this paper first summarizes the experience of global anti-epidemic work in the past two years since the outbreak of COVID-19, analyzes CEPC from different dimensions and perspectives, summarizes the research methods and technical tools of CEPC and identifies the problems of CEPC in China. Then, based on the dynamic network theory, the community anti-epidemic dynamic network and anti-epidemic resilience evaluation index system are constructed by combining the existing studies. Finally, a simulation platform is designed through computational experiments, and a multivariate community resilience strategy optimization model is established. Finally, it responds to the requirements of CEPC and makes a theoretical contribution to the world anti-epidemic work.

## 2. Research Progress

Communities in China have experienced nearly a hundred years of development and evolution. With the promotion of China’s social transformation, “community governance” has gradually replaced the concept of “community management”, and the content of governance has gradually expanded from daily health rectification, grassroots management, party–government relations and ecological civilization construction to issues with contemporary characteristics, such as disaster and emergency prevention and control, smart community construction, old community renovation and aging population coping strategies. The governance perspective has also undergone a transformation process from “social systems theory” to “demand theory” to “resilience, empowerment and advantage theory” [9].

The concept of resilience is used in many fields, such as ecosystems, social systems and socio-ecological systems, to denote “the ability of a system to withstand external shocks and maintain its primary structure and functions in the event of a crisis” [10]. Applying resilience to CEPC can make up for the limitations of the “top-down” prevention and control management system in China. Compared to traditional communities, resilient communities have the “4R”characteristics of Robustness, Redundancy, Rapidity and Resourcefulness and tend to suffer less and recover faster from disasters in the face of the same shocks and stresses [11]. Since the establishment of the Action Program of “Improving the Resilience of Nations and Communities to Adapt to Disasters“ at the Second World Conference on Disaster Reduction of the United Nations, countries worldwide have successively launched explorations on building resilient communities. The establishment of resilient communities is an important way to enhance community resistance to epidemics and optimize the national public health emergency prevention and control system of China.

### 2.1. Ideas and Perspectives on CEPC

Some studies have decomposed the community into different dimensions from the perspective of resilience and analyzed the CEPC in terms of various aspects under public health emergencies—for example, measuring the level of anti-epidemic resilience in four areas: subject capacity development, networked participation, consolidation of social capital and focus on digital intelligence governance [5]; analyzing the path of community anti-epidemic capacity construction from physical, economic, social and institutional dimensions [12]; improving the community’s anti-epidemic resilience from three levels of function, system and practice [13]; building community anti-epidemic resilience from three dimensions: resilience subject, resilience model and resilience goal [14]; starting from the stakeholders of public institutions, community residents, social organizations and volunteers, and applying the vulnerability analysis matrix for epidemic resilience assessment [15]; proposing the resilient community creation strategy from the three perspectives of facility and space hardware guarantee, governance service resource matching and governance capacity system construction [16].

From the perspective of multi-subject collaborative governance, some scholars aim to achieve the emergent effect of CEPC through efficient collaboration among community anti-epidemic stakeholders. For example: connecting the community with the government, medical institutions, social forces and residents through the three major networks of information, decision making and service to form a community public health emergency coordination network system [15,17]; using the “directive execution” model in dealing with community public health emergencies [18]; optimizing the linkage of communities, social organizations and social workers from institutional empowerment, technical empowerment and normal and emergency coordination [19]; enhancing the effectiveness of CEPC through community social capital and government grading guidance [20]; building a more effective network of community actors will promote the transformation and upgrading of community governance [21].

Some studies consider communities as the source of epidemic prevention and control and start from a grassroots governance perspective to strengthen the construction of professional teams, emergency response capacity, resource integration capacity, information application capacity and relevant regulations and systems at the grassroots level [22]. At the same time, construct and form an emergency mechanism or platform led by or involving the participation of community work so that the advantages of community work can be fully utilized [23]. Some studies have pointed out that professional community managers should go down to the grassroots community to construct a grassroots community governance system integrating “autonomy, rule of law and rule of morality” [24]. From the perspective of system construction and innovation, some previous studies have emphasized the importance of strengthening residents’ participation [25] and carrying out epidemic prevention exercises [26].

### 2.2. Theoretical Framework and Indicators of CEPC

By establishing theoretical frameworks, previous studies can clarify the research ideas of community anti-epidemic work and lay a theoretical foundation for subsequent evaluation and strategy research. For example, the Epidemic Management Framework provides guidance for managing the four phases of outbreak preparedness, response, recovery and mitigation through automated content identification and analysis of COVID-19 reports in local newspapers [27]; the Outbreak Management Framework and its application tools provide data support for controlling the spread of the epidemic by tracing contacts in specific areas [28]; the *4R* Crisis Management Framework proposes to build a “point, line, surface, body” method to improve the CEPC capabilities [29]; the 6*Cs* Conceptual Framework states that the cognition, communication, collaboration, control, confidence and co-production are critical to disaster response [30]; the Multi-actor Coordination Framework models the coordination of activities, actors and resources in an outbreak management scenario, identifying 25 key resources and eight important activities for epidemic management [31]; the *W2R* Framework includes recommendations for community management in the three areas of “pre-epidemic health risk warning, mid-epidemic planning and building response and post-epidemic community revitalization” [32].

To identify key CEPC factors, some scholars have studied the quantitative or qualitative relationship between influencing factors and the capacity of community resilience. For example, through Grounded Theory, identifying the factors as a result of which urban residents fail to respond to the governance of the community epidemic in a timely manner and the mechanism of action between them [33], or recognizing the influencing factors (political, economic, socio-cultural, infrastructure and human health) of community epidemic recovery [34]; validating the relationship model between community awareness, community resilience and mental health through Structural Equation Modeling [35]; using the *Pearson Correlation Coefficient* to characterize the correlation between the number of confirmed cases of COVID-19 in Wuhan and socioeconomic factors such as community characteristics and distance variables [36]; the impact of the level of urban governance on CEPC and the mental health risk of residents is analyzed by Ordinary Least Squares [37,38,39]; using Multiple Linear Regression to analyze the main influencing factors and influence degree of community resilience during the epidemic [5]; establishing a Relative Importance Matrix to analyze the relationship between community residents’ satisfaction and local government performance during the COVID-19 epidemic [40]; using Sensitivity Analysis to evaluate the impact of key influencing factors on the epidemic contribution of disaster response capability [41].

### 2.3. Research Methods and Technical Means for CEPC

The research methods related to CEPC can be categorized as research interview, mathematical model, computer-based learning and others. These methods are developed to collect elements, information and data related to CEPC, to identify, analyze and respond to risks or to establish a management platform based on big data. According to the above research results, the CEPC strategies are formulated, as shown in Table 1.

### 2.4. Commentary of Previous Studies

To sum up, the existing studies have enriched the content of urban community resilience governance and public health emergency management, as well as provided ideas and inspiration for the governance of CEPC in China. However, the previous studies in this area cannot form a comprehensive theoretical and technical support, as follows:(1)Analysis of the interaction between key elements such as information and epidemic disturbances does not have a deep understanding of the definition of community anti-epidemic resilience.(2)Existing studies have attempted to classify the key factors of CEPC in terms of community resources, activities and social capital. However, there are few quantitative analyses between “incoming causes” (inspired by the disturbance of the epidemic) and the “outgoing effects” (the impact path of the anti-epidemic network), which makes it difficult to reveal the evolutionary mechanisms of urban community anti-epidemic network changes and anti-epidemic resilience.(3)Some studies use multiple linear regression, path analysis, the fuzzy comprehensive evaluation method, the analytic hierarchy process and cluster analysis to classify, quantify and measure the level of epidemic prevention and control in urban communities. However, in the post-epidemic era, traditional static assessment methods ignore the changes in community resilience dynamics and the impact of subject behavior on resilience, and they are no longer applicable to the complex, dynamic and evolving overlapping CEPC efforts.(4)Studies have been conducted to propose optimal strategies for epidemic prevention and control in urban communities from different perspectives such as organizational structure optimization, institutional construction and physical facility strengthening. However, most of them are not supported by prior quantitative evaluation results and data, and less consideration is given to the impact of changes in epidemic disturbance scenarios and community cost constraints on strategy formulation. This resulted in the lack of intelligence, objectivity, timeliness and relevance of the formulated policy advice.

In response to the problems existing in the CEPC work in China, and the goal of “strengthening the construction of China’s urban CEPC system”, the following measures have been formulated: (1) build a composite network including people, information, tasks and resources to describe the normal state and structure of the community’s anti-epidemic work; (2) introduce disturbance variables, act them on the network to make it evolve and, finally, reach a new balance so as to describe the situation changes in the community after being affected by the epidemic; (3) select the key parameters in the dynamic network to measure the resilience of the network at different stages and to describe the ability of the CEPC; (4) establish an experimental model of simulation calculation, and output different resilience and corresponding anti-epidemic strategies by adjusting the community epidemic situation and cost input.

## 3. Improvement Ideas

### 3.1. Community Anti-Epidemic Dynamic Network and Anti-Epidemic Resilience Evaluation Index System

First, according to the characteristics of epidemic spreading in urban communities in China, the physical and process boundaries of CEPC are determined. This is carried out through multi-source data collection using web crawler technology, based on the “Houyi Collector” software, with “community + outbreak” as the search keywords to count the reports of the community epidemic since the outbreak of COVID-19 in December 2019. Then, visits, questionnaires, in-depth interviews and other forms are used to supplement the case data. Eventually, the typical events of CEPC in China are classified and coded according to different attributes such as scale, duration, spatial distribution, number of patients and governance effects, and an urban community epidemic case database is built.

Second, based on the dynamic network theory, extract the key element information in the case, set up multi-layer network element matrix nodes and the relationship between nodes and build an anti-epidemic network under a normal state, including four types of nodes including stakeholders (S), resources (R), information (I) and tasks (T). For example, S nodes include community neighborhood committees, residents, street offices, voluntary service organizations, etc.; R nodes include medical supplies, food, location advantages, etc.; I nodes include epidemic information online, government instructions, resident reports, etc.; T nodes include COVID-19 testing, isolation, vaccination, publicity, registration, etc. Then, after weighting the relationship between nodes by the Delphi method, four sub-networks, including the stakeholder network (SS), resource network (SR), information network (SI) and task network (ST), composed of the above nodes are obtained, as shown in Figure 1.

According to the 4 Rs of Resilience, each “R” can be used to represent an aspect of the resilience of a community network. The first “R” is robustness, which is mainly expressed by the centrality indicator. The higher the centrality, the stronger the relationship between stakeholders and the more solid the network of social subjects; thus, the robustness can be used to characterize the resilience of the subject network. The second “R” is redundancy, which is mainly described by the structural hole in the Social Network Analysis (SNA). The stronger the redundancy of a community, the fewer structural holes it has; then, redundancy can be used to characterize resource network resilience. The third “R” is rapidity, which is mainly expressed by community cohesion. The higher the density in the community social network, the shorter the communication path between stakeholders, and more timely information can be obtained to respond to the situation in which an epidemic strikes, so rapidity is chosen to characterize information network resilience. The fourth “R” is resourcefulness, which is mainly shown by small group analysis. The stronger the small group, the more resourceful the community is, and the more options stakeholders can choose to accomplish the task of fighting the epidemic, so it is appropriate to use resourcefulness to characterize task network resilience.

Centrality degree/transitivity indicators can be used to characterize robustness as a way to describe and show the power and reputation of stakeholders and the mediating and information bridging functions in a community during an epidemic. From the perspective of the whole network, a structural hole can be described as an empty hole in the network structure [58]; it is not directly connected to other individuals when it is directly connected to one or some individuals in the social network. Based on Burt’s structural hole index [59], redundancy is expressed in this paper by the scale/efficiency/constraint. Efficiency is the effective scale of points divided by the actual scale of points in a single network, and it is used to describe the degree of influence of a node on other related nodes in the network. Constraint indicates the closeness of a single network, which is the degree to which a node in the network is directly or indirectly close to other nodes. The network density is used to measure the degree of cohesiveness of the community. The more ties among stakeholders, the greater the network density and the influence on participants’ attitudes and behaviors. The cohesive subgroup indicator in small group analysis is used to measure resourcefulness in community network resilience. When certain participants in a network are particularly close to each other to the extent that they trust each other and are merged into a small group, this is referred to in SNA as a cohesive subgroup based on reciprocity.

Further, based on resilience theory, community governance theory and the 4R characteristics of resilience, the concept of resilience of urban communities against epidemics is defined. In general, it includes using (1) robustness to characterize the resilience of the SS network, (2) redundancy to the SR network, (3) rapidity to the SI network (4) and resourcefulness to the ST network. Some key indicators of the dynamic network are selected to characterize the level of resilience of different aspects, including centrality (centrality degree/transitivity), structural hole (scale/efficiency/constraint), cohesion (network density) and cliques (cohesive subgroups), respectively.

Grounded Theory is a bottom-up procedural qualitative research method that emphasizes inductive guided analysis of a phenomenon based on empirical data, with an open mind and developing theory in continuous generalization. The application process of grounded theory can be divided into the following seven stages: research question formulation, a priori knowledge review, data collection and organization, data induction and coding, preliminary theory building, theory saturation testing and research conclusion generation. It can be seen that the rooting theory is particularly suitable for studies related to the lack of theoretical explanations or the inadequacy of existing explanations. Therefore, on the one hand, based on the Grounded Theory, conduct in-depth interviews with stakeholders such as sub-district offices, neighborhood committees, community properties and community residents, and convert the collected audio, video and other materials into text, coding and systemization; on the other hand, based on the literature review method, the key words “community + epidemic + resilience” were used to collect and select the indicators in the mainstream databases such as WOS, Elsevier, Wiley online and China Knowledge Network to build the evaluation index system of urban community anti-epidemic resilience, as shown in Figure 2.

### 3.2. Evolutionary Mechanism and Evaluation Method of Anti-Epidemic Resilience

First, the epidemic disturbance variables are introduced, including stakeholder disturbance, resource disturbance, information disturbance and task disturbance. Based on the coupling theory and grey system theory, the coupling probability, weight and coupling degree on the network among the disturbance factors are calculated. Then, the coupling effect of the epidemic disturbance factors is analyzed, and the coupling set of the urban community epidemic disturbance factors is established. Second, the development process of anti-epidemic in urban communities is divided into three stages: “normal state—emergency state—recovery state”, and the end of the recovery state marks the community’s entry into a new normal state of anti-epidemic. Thirdly, through Bayesian theory, a vulnerability-node excitation model of the urban community’s anti-epidemic resilience dynamic network is established, the disturbance factors are acted on the vulnerable nodes and the induced paths and functional relationships of the disturbance factors are clarified. Finally, the evolution of the 4R characteristics of resilience in the process of network changing is observed, including indicators such as the degree of disturbance, duration, node relationship and overall network performance, until the network reaches a new dynamic balance, as shown in Figure 3.

Second, establish a community anti-epidemic structural equation model, and collect, count and test the data of the anti-epidemic resilience and disturbance factors through the database and survey interviews to determine the qualitative relationship among indicators. Furthermore, the deep learning is used to input and train the historical data of community anti-epidemic resilience. Then, determine the quantitative relationship between community anti-epidemic resilience and disturbance factor indicators based on the functional relationship and algorithm. The accuracy of the above qualitative and quantitative relationships is verified by the reserved data test.

Then, the community resilience is evaluated with a “full element participation, whole process evolution and multi-dimensional dynamic interaction” perspective. The evaluation consists of two criteria: criterion one is based on the multi-objective decision making and integrated evaluation method, which measures the level of a community’s anti-epidemic resilience by inputting the network parameters of the target community; the second criterion is based on the classic Deep Belief Network (DBN) model. The index data are simulated in the DBN network, and the potential anti-epidemic resilience of the community under a certain state in the future is output by inputting the normal anti-epidemic network parameters, as shown in Figure 4. Where H, V and W represent the hidden units, visible units and weights, respectively.

As a deep learning algorithm, DBN is widely used for big data prediction, data mining, identification and classification [60]. The DBN model is composed of a multi-layer restricted Boltzmann machine (RBM) and a one-layer backpropagation network (BP) stack. Compared with a single RBM, multiple RBMs of a DBN model can further extract deep features of complex data, so the DBN model has a powerful learning capability [61]. Moreover, compared with the traditional artificial neural network, DBN adopts an unsupervised pre-training method, which greatly enhances the data mining ability and improves the prediction accuracy [62]. Therefore, using the DBN model to train community resilience data indicators and using it to predict the resilience level of the community will have a high accuracy.

### 3.3. Output of Strategies to Improve the Communities’ Anti-Epidemic Resilience in China

First, build the computational experiment platform and embed the modules required for platform operation, mainly including: (1) Dynamic Bayesian network module. Use the dynamic Bayesian network to build a community anti-epidemic resilience evolution model, define attributes of network nodes, relationship chains and epidemic disturbance factors in the AnyLogic platform and quantify the relationship among elements through embedded functions and algorithms; (2) Multi-agent simulation module. Construct a multi-intelligence simulation model with stakeholders as the main body, and map the actual situation of organizational structure, authority and responsibility tasks, resource deployment and information transfer among stakeholders to the virtual environment. (3) Dynamic parameter settings. Integrate the multi-agent simulation module with the dynamic Bayesian network module on the AnyLogic platform, and set parameters such as subject attributes, disturbance scenarios, situation timing and network link disturbance paths; (4) Multi-source data integration module. It includes building the model-running environment, setting the input/output data of the simulation model, coding resilience measures, setting disturbance scenarios and cost constraint functions, designing informational expressions and visualization interfaces, calling database interfaces and so on; (5) Parallel experiment module. The interface is reserved for accessing the anti-epidemic case database of urban communities, updating the latest community epidemic information, inputting it to the simulation platform for collaborative evolution and providing feedback to correct the resilience evolution and measurement model.

Second, cluster analysis is conducted on the resilience level of community anti-epidemic, and the resilience level is divided into several levels to determine the resilience threshold under different epidemic situations. Then, the key parameters of the dynamic network are set as adjustable variables, and the alternative sets of “perturbation scenarios–optimization strategies” for different levels of resilience are constructed based on scenario derivation and gray system theory. Further, the anti-epidemic cost constraint variable is introduced to consider the actual cost expenditure of the target community. Based on the measurement criteria, a multi-objective optimization model of community resilience enhancement strategy is established with the objective function of “maximizing resilience + minimizing input cost”, and the strategy set of “epidemic disturbance—cost constraint—anti-epidemic resilience” is output. The platform architecture and operation process are shown in Figure 5.

## 4. Conclusions

This paper summarized the current research progress of urban community anti-epidemic resilience from three perspectives: the ideas and perspectives, theoretical framework and index system and research methods and technical means. It was found that the resilience concept has a future in the field of community epidemic or public health emergency prevention and control. The current research has gaps such as an unclear definition of the anti-epidemic resilience, an unclear evolution mechanism and imprecise evaluation methods.

Consider that the CEPC work is a multi-dimensional dynamic network involving many factors, such as people, information, resources, events and uncertain disturbances, and is constantly changing over time. Therefore, this paper proposed the integration of the dynamic Bayesian network, deep belief network, multi-agent simulation and other technical means from the perspective of dynamic network analysis and deconstructed the complex CEPC work into several networks coupled with each other. By analyzing the evolution of dynamic networks and mapping reality to virtual reality, this paper searched for weak points in the whole process of CEPC. On this basis, key indicators of resilience were selected as the evaluation criteria to measure the level of CEPC in the process of the situation changing. Finally, a simulation platform was designed through computational experiments, and a multivariate community anti-epidemic strategy optimization model was established. In tandem with parallel experiments, the combination of imagination and reality and feedback correction, the purpose of the intelligent output of the resilience level and the intelligent optimization of strategy was realized and responded to the requirements for CEPC.

The improvement ideas of anti-epidemic resilience presented in this paper provide an effective reference for other countries in improving the anti-epidemic resilience in urban communities. Nevertheless, there is still much work to be done in the future. When measuring the level of CEPC during the situation change, some other key indicators and more universal resilience indicators can be chosen as the assessment criteria. In addition, this paper only provided an idea for the improvement of anti-epidemic resilience in China, and future studies can consider using some actual data to demonstrate the simulation results of modeling as empirical examples and to discuss the accuracy and reliability of model predictions. Moreover, China is a huge country, and there are great differences between different cities. This paper does not explicitly consider the differences between cities. Therefore, this needs to be taken into account in future studies to solve the different problems existing in different urban communities.

## Figures and Tables

**Figure 1 ijerph-19-15293-f001:**
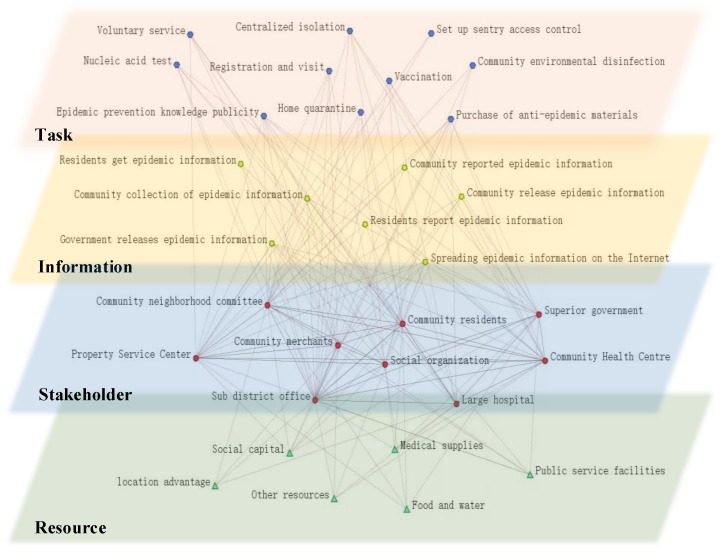
The anti-epidemic network and relationships in urban communities under a normal state.

**Figure 2 ijerph-19-15293-f002:**
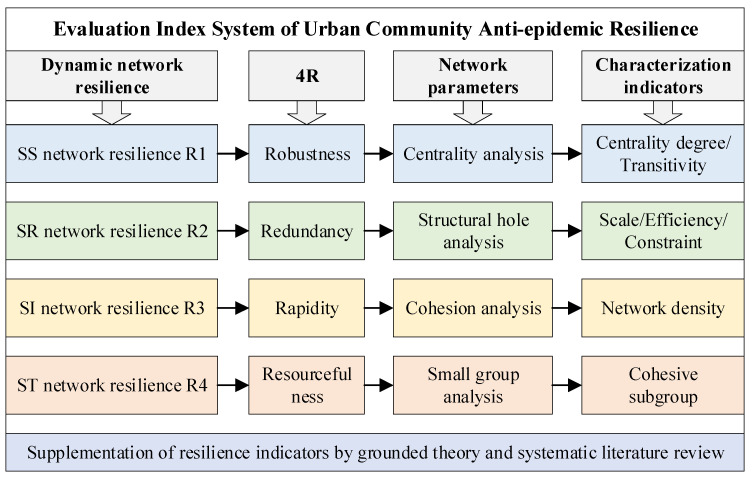
Flow chart of the construction of an urban community resilience evaluation index system.

**Figure 3 ijerph-19-15293-f003:**
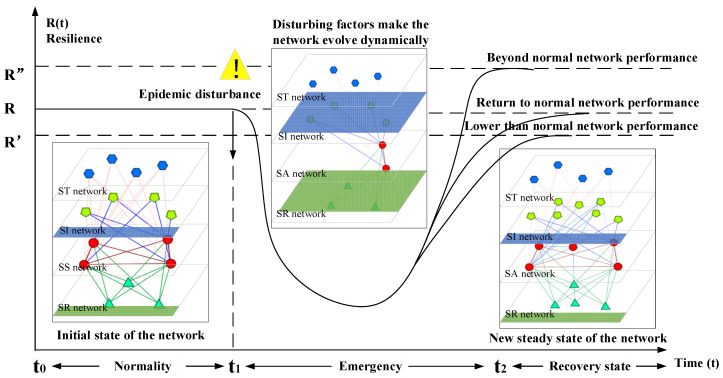
Schematic diagram of the dynamic network disturbance mechanism of urban community anti-epidemic.

**Figure 4 ijerph-19-15293-f004:**
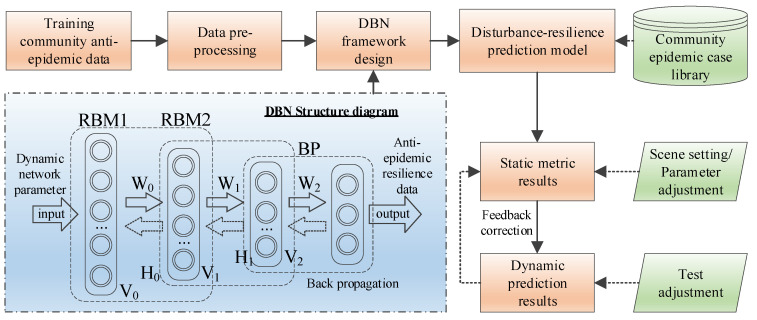
Forecast flow chart of community anti-epidemic resilience.

**Figure 5 ijerph-19-15293-f005:**
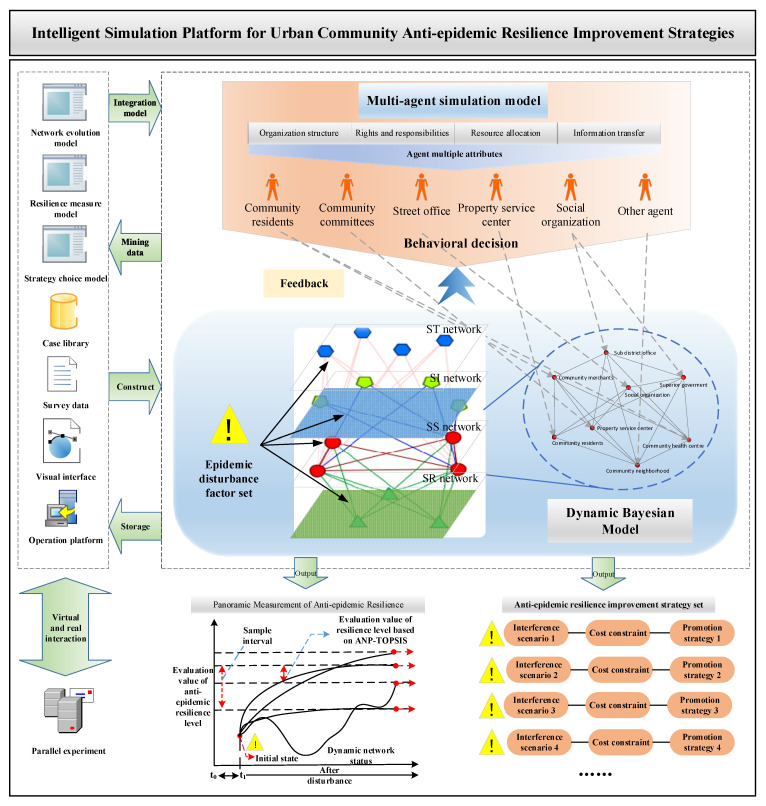
Intelligent simulation platform for urban community anti-epidemic resilience improvement strategies.

**Table 1 ijerph-19-15293-t001:** Review on research methods of CEPC.

Type of Methods	Research Methods	Main Research Objectives	Sources
Research interview	Social media public opinion analysis	Analyze the implementation of community emergency management projects under public health emergencies	[42]
		Build an all-around and full-process “Internet +” CEPC platform	[43]
	Community informatics	Identify false information/misinformation and improve community public health emergency preparedness and response	[25]
		Collect and analyze community information and data to formulate grassroots epidemic prevention and control strategies	[44]
		Reasonable use and protection of the digital information of community residents to provide strategies for community governance	[45]
	Narrative description	Establish and implement emergency response workflows in response to the epidemic outbreak	[46]
	Research interviews	Conduct field surveys in communities to maximize the use of local community infrastructure	[47]
	Interactive ceremony chain	Explore the key factors affecting the effectiveness of CEPC in the public health governance space embedded in “ceremonies”	[48]
Mathematical model	Hatton matrix + CRR combination model	Identify factors contributing to epidemic susceptibility and severity, and propose prevention and mitigation strategies	[49]
	Pressure-state- response model	Build an urban CEPC data ecological information governance system and identify the risks	[50]
	Birthday paradox probability model	Establish a community complex network system social risk assessor to assess community exposure risk during the current epidemic	[51]
	Improved SEIR infectious disease model	Assess the effectiveness of epidemic control measures such as closures, movement restrictions, social distancing, etc.	[52]
	Truncated regression model	Investigate whether epidemic prevention measures and characteristics of the community affect residents’ awareness	[53]
Computer-based learning	Kernel density estimation + Geographically weighted regression	Explore spatial differences in epidemic intensity and quantify the impact of population dynamics, transportation and social interactions on epidemics	[54]
	SEIR model + K-means cluster analysis	Analyze global community-acquired epidemic patterns to support epidemic surveillance and non-drug interventions	[55]
	Case-based reasoning	Construct a community epidemic management framework consisting of the key elements of basic and support activities	[56]
	Quick reference handbook concept	Establish a community risk management system for infectious diseases by compiling a guidebook for community residents to respond to disasters	[57]

## Data Availability

The data presented in this study are available on request from the corresponding author. The data are not publicly available due to privacy.

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
