# Peer review of "Research Progress and Improvement Ideas of Anti-Epidemic Resilience in China’s Urban Communities"

_ijerph, 2022, doi:10.3390/ijerph192215293_

Round 1

Reviewer 1 Report

This paper is addressing the problem of anti-epidemic resilience in China’s urban communities. The topic of the paper is interesting and valuable in the field of constructing resilient communities. In general, this study was carefully organized, but some parts need to be strengthened before publication.

1. In “Introduction” section, almost all data statistics did not indicate the sources. Please check and indicate the references.

2. At the beginning of the “Introduction”, the authors should first introduce the major epidemic prevention status and the community's performance in epidemic prevention all over the world, and then introduce the situation in China.

3. The authors can describe how the manuscript is organized at the end of “Introduction” part.

4. Lines 68-69, page 2, “Numerous cases of community outbreaks demonstrates that many management deficiencies of CEPC exist in the normal and emergency work in China. For example…”. It is important for the authors to provide references here.

5. The authors should introduce how the study was conducted at the end of “Research Progress” part.

6. The readability of Figure 1 is poor, such as font and content.

7. Based on the framework proposed by the authors, how to prove its effectiveness? Or is the framework easy to implement? If possible, please provide cases or evidence.

8. The authors have not yet discussed the shortcomings and limitations in current writing (proposed framework/details). Please add it in new section “Discussion and limitations”.

9. Please simplify the “Conclusions” section. The “Conclusions” part should clearly indicate the main conclusions of the study.

Reviewer 2 Report

This article is classified as a review, however, the methodology used to search for and select the articles included is not described in detail. 

On the other hand, the conclusions are too extensive and repetitive in relation to the information previously presented.

For all the above reasons, I consider that this article requires a thorough revision of these methodological aspects. 

Reviewer 3 Report

I appreciate authors’ efforts on reviewing the literature, presenting their ideas/frameworks on anti-epidemic resilience among China’s urban communities, and illustrating the simulation platform in the context of COVID-19 pandemic.

I feel it would be more valuable if authors could use some actual data to show their model/platform simulation results as an empirical example, and discuss the results (e.g. accuracy, reliability of the predictions, etc.).

Additional discussion on actual implementation and policy implication would also be helpful. China is a huge country, and I would expect large differences across different cities. How could authors' framework and simulation platform address unique issues across different urban communities in China? Again, some concrete examples would be helpful. 

Round 2

Reviewer 1 Report

The authors have revised this manuscript as required, and it is now acceptable in current writing.